# Ubiquitin-Proteasome Modulating Dolabellanes and Secosteroids from Soft Coral *Clavularia flava*

**DOI:** 10.3390/md18010039

**Published:** 2020-01-03

**Authors:** Che-Yen Chiu, Xue-Hua Ling, Shang-Kwei Wang, Chang-Yih Duh

**Affiliations:** 1Department of Marine Biotechnology and Resources, National Sun Yat-Sen University, Kaohsiung 80441, Taiwan; m025020014@student.nsysu.edu.tw (C.-Y.C.); cooley@mail.nsysu.edu.tw (X.-H.L.); 2Department of Microbiology and Immunology, Kaohsiung Medical University, Kaohsiung 80708, Taiwan

**Keywords:** proteasome inhibition, dolabellane, secosteroids, soft coral

## Abstract

We performed a high-content screening (HCS) assay aiming to discover bioactive molecules with proteasome inhibitory activity. By structural elucidation, we identified six compounds purified from soft coral *Clavularia flava*, which potentiates proteasome inhibition. Chemical structure elucidation revealed they are dolabellane- and secosteroid-based compounds including a new dolabellane, clavinflol C (**1**), three known dolabellanes, stolonidiol (**2**), stolonidiol-17-acetate (**3**), and clavinflol B (**4**) as well as two new secosteroids, 3*β*,11-dihydroxy-24-methyl-9,11-secocholest-5-en-9,23-dione (**5**) and 3*β*,11-dihydroxy-24-methylene-9,11-secocholest-5-en-9,23-dione (**6**). All six compounds show less cytotoxicity than those of known proteasome inhibitors, bortezomib and MG132. In summary, the high-content measurements of control inhibitors, bortezomib and MG132, manifest the highest ratio >2 in high-content measurement. Of the isolated compounds, **2** and **5** showed higher activity, followed by **3** and **6**, and then **1** and **4** exhibited moderate inhibition.

## 1. Introduction

Marine natural products harbor unique chemical structures and exhibit diverse biological activity with potential therapeutic utilities that merit investigation [1]. Synthetic drugs with proteasome inhibition, Velcade (bortezomib), Kyprolis (carfilzomib), and Ninlaro (ixazomib), exemplify therapeutic efficacy for the treatment of multiple myeloma [2,3]. In an effort to develop target-direct drug screening assay, we validated a sensitive and efficient high-content screening assay for discovery of proteasome inhibitor. Our initial efforts identified four natural products of soft corals cembrane-based compounds (sarcophytonin A, sarcophytoxide, sarcophine, and laevigatol A) which potentiate proteasome inhibition [4]. We postulate that proteasome inhibitors may benefit the ecosystem of soft coral reef holobiont. Therefore, we continued the drug screening in an effort to identify marine natural products purified from Formosan soft corals in our laboratory. In this study, we demonstrated the identification of six compounds with proteasome inhibitory activity. Structure elucidation revealed they are dolabellane- and secosteroid-based compounds including a new dolabellane, clavinflol C (**1**), three known dolabellanes, stolonidiol (**2**), stolonidiol-17-acetate (**3**), and clavinflol B (**4**) as well as two new secosteroids, 3*β*,11-dihydroxy-24-methyl-9,11-secocholest-5-en-9,23-dione (**5**) and 3*β*,11-dihydroxy-24-methylene-9,11-secocholest-5-en-9,23-dione (**6**) (Figure 1).

## 2. Results

### 2.1. Compound Purification and Structure Elucidation

Chromatographic fractionation of ethyl acetate solubles from *C. flava* afforded four dolabellane diterpenes, **1**–**4**, as well as two secosteroids, **5** and **6**. The three known dolabellane diterpenes, **2**–**4** were identified by comparison of their spectral data with those of reported literatures [5,6]. The structures of new compounds, **1**, **5**, and **6** were elucidated by analysis of 1D and 2D spectral data (Appendix A).

Clavinflol C (**1**) had a molecular formula of C_20_H_33_O_4_Cl as deduced from HR-ESI-MS (Appendix A) and NMR data. Its IR bands (Appendix A) indicated the presence of exo-methylene (1,639, 961 cm^−1^) and hydroxyl (3343 cm^−1^) groups. The ^1^H NMR data of **1** (Appendix A) showed a pair of exo-methylene singlets (δ 4.91, 5.00) and an AB quartet for hydroxy-methyl group (δ 3.39, 3.82, J = 11.6 Hz), three methyl singlets (δ 1.03, 1.32, 1.40), an oxygenated methine proton (δ 4.18), and a chlorinated methine proton (δ 4.00). To determine the proton sequence of **1**, a COSY spectrum (Appendix A) revealed the connectiveness of H-2/H-3, H-5/H-6/H-7, H-9/H-10, and H-13/H-14. The ^13^C NMR (Appendix A) and HSQC spectra (Appendix A) of **1** showed signals for three methyl carbons, eight methylene carbons including the exomethylene (δ 113.9, 147.7), two methine carbons, and five quaternary carbons. Detailed analyses of the ^1^H, ^13^C NMR, and HSQC spectra revealed that **1** is a dolabellane diterpene with a 5/11 membered ring and a tetrasubstituted olefin at the C-11/C-12 positions. This type of skeleton was further confirmed from the observation of long range correlations of H_2_-16/C-3, C-5; H_2_-2/C-4; H_2_-7/C-8, C-17; H-10/C-1, C-7, C-8, C-9, C-11, H_3_-15/C-1, C-2, C-11, C-13; H_2_-13/C-1, C-11, C-12; H_2_-14/ C-11, C-12; H_3_-19/C-12, C-18, C-20; H_3_-20/C-12, C-18, C-19 in the HMBC spectrum (Appendix A). The relative stereochemistry of **1** was determined from NOESY experiments as illustrated in Appendix A. Assuming that H_3_-15 is α-oriented, key NOESY correlations from H_3_-15 to H-10 and from H-7 to H-10 suggested that H-10 and H-7 were in the α-orientation. NOESY correlations between H-9a/H-6a and H-9b/H-17b suggested that OH-8 was in the *β*-orientation.

Compound **5** was isolated as a white amorphous powder, showing a pseudo-molecular ion peak at *m*/*z* 469.32880 [M + Na]^+^ in the HR-ESI-MS (Appendix A), consistent with the molecular formula C_28_H_46_ NaO_4_ (calculated for 469.32883), requiring six degrees of unsaturation. The presence of an oxymethylene and a keto carbonyl carbon was confirmed by the ^1^H NMR (Appendix A) (δ_H_ 3.88 (m, H-11a) and 3.74 (m, H-11b)) and ^13^C NMR (Appendix A) (δ_C_ 59.2 (CH_2_), 212.5 (qC), and 216.2 (qC)) data, as well as from the IR absorption (Appendix A) at 3396 and 1704 cm^−^^1^. The diagnostic NMR signals of a 9,11-secosterol were confirmed by the ^1^H–^1^H COSY correlation (Appendix A) from H_2_-11 to H_2_-12 as well as HMBC correlations (Appendix A) from H3-18 to C-12, C-13, C-14, and C-17; from H3-19 to C-1, C-5, C-9, and C-10. The NMR features of **5** were analogous to those of 3,11-dihydroxy-24-methyl-9,11-secocholest-5-en-9-one [7], except for the presence of a ketone (δ_C_ 201.1 (qC)) at C-23. Based on NOESY correlations (Appendix A) of H_3_-19/H-1, H_3_-19/H-2, H_3_-19/H-4, H_3_-19/H-8, H-3/H-1, H-3/H-2, H-3/H-4, H-8/H_3_-18, H-8/H-7, H_3_-18/H-15, H_3_-18/H-16, H_3_-18/H-20, and H-14/H-7, the relative stereochemistry at C-3, C-8, C-10, C-13, C-14, C-17, and C-20 in **5** was found to be the same as those of 3*β*,11-dihydroxy-24-methyl-9,11-secocholest-5-en-9-one [7]. On the basis of the above-mentioned findings, the structure of **5** was consistent with the structure shown as 3*β*,11-dihydroxy-24-methyl-9,11-secocholest-5-en-9,23-dione.

Compound **6** appeared as a white amorphous powder like **5**. Careful inspection of the 2D NMR spectroscopic data (Appendix A) of **6** led to the establishment of the same nucleus as that of **5**. The NMR spectroscopic data (Appendix A) of **6** were analogous to those of **5**, except for NMR signals due to the conjugated enone in **6**. The location of the conjugated enone was identified by the HMBC correlations (Appendix A) from the methylene protons (H_2_-22) to the carbonyl carbon (C-23) and from H_3_-26, 27 to C-24, securing the structure of **6**, which was shown as 3*β*,11-dihydroxy-24-methylene-9,11-secocholest-5-en-9,23-dione.

### 2.2. Identification of Marine Compounds Showed High-Content Characteristics of Proteasome Inhibition

The proteasome inhibition assay was performed by following the standard operation protocol of high-content screening (HCS) of EGFP-UL76 aggresome as described previously [4]. A stringent proteasome inhibition was considered as the HCS measurements of marine compounds with an increase greater than 0.2-fold relative to those of the control without treatment. Under this validity criterion, we demonstrated the identification of six compounds with proteasome inhibition and their effects were statistically significant. Four compounds with dolabellane-based structures designated clavinflol C (**1**), stolonidiol (**2**), stolonidiol-17-acetate (**3**), and clavinflol B (**4**) (Figure 2 and Figure 3) [5,8]. Additionally, two unprecedent compounds with secosteroid-based structures designated compound **5** and **6** (Figure 4 and Figure 5). Prior to HCS experiments, the in vitro cell-based MTT (3-(4,5-Dimethylthiazol-2-yl)-2,5-diphenyltetrazolium bromide) cytotoxicity assays were performed against four cell lines: A549 (human lung adenocarcinoma), HT-29 (human colon adenocarcinoma), and P-388 (mouse lymphocytic leukemia). Plasmid pEGFP-UL76 transfected HEK293T (human embryonic kidney large-T antigen-transformed) cell expressing EGFP-UL76 for HCS assay was assessed the ED_50_ using both MTT and high-content nuclear count measurements (Appendix A). The ED_50_ values for respective compounds were as follows: compound **1**, >50 μg/mL, >50 μg/mL, >50 μg/mL, >25 μg/mL, and 6.14 μg/mL; stolonidiol (**2**), 3.9 μg/mL, >50 μg/mL, 0.6 μg/mL, >25 μg/mL, and >25 μg/mL; stolonidiol-17-acetate (**3**), >50 μg/mL, >50 μg/mL, >50 μg/mL, >25 μg/mL, and 19.56 μg/mL; clavinflol B (**4**), >50 μg/mL, >50 μg/mL, >50 μg/mL, >25 μg/mL, and 21.33 μg/mL; compound **5**, >50 μg/mL, 3.2 μg/mL, 4.6 μg/mL, 12.28 μg/mL, and 12.13 μg/mL; compound **6**, 5.3 μg/mL, >50 μg/mL, 4.8 μg/mL, >25 μg/mL, and 10.92 μg/mL. Both the MTT assay and high-content nucleus counts were performed to assess the ED_50_ values of HEK293T cells expressing EGFP-UL76 for bortezomib which were 11.95 nM and 24.29 nM and for MG132 were 1.18 μM and 1.91 μM, respectively. Clavinflol B (**4**) showed moderate cytotoxicity in previous reports, which was consistent with our data (Appendix A) [6].

Following the HCS assay, the high-content EGFP-UL76 aggresome measurements integrated intensity and average intensity per cell were analyzed and the relative ratios were obtained by normalization to the control. For the ratio of EGFP-UL76 aggresome integrated intensity per cell (Figure 2A and Figure 3A, top panels), the highest ratios for compounds **1**, **2**, **3**, **4**, **5,** and **6** were 1.22 (*p* = 0.0390), 2.12 (*p* < 0.0010), 1.74 (*p* = 0.0020), 1.33 (*p* < 0.0010), 2.03 (*p* < 0.0010), and 1.72 (*p* < 0.0010), respectively. The highest ratios of average intensity per cell presented for compounds **1**, **2**, **3**, **4**, **5,** and **6** were 1.32 (*p* = 0.0371), 1.75 (*p* = 0.0021), 1.40 (*p* < 0.0010), 1.19 (*p* = 0.0117), 1.85 (*p* < 0.0010), and 1.34 (*p* = 0.0089), respectively (Figure 2A and Figure 3A, bottom panels). Furthermore, all these increases in ratios achieved statistical significance.

Consequently after the assay procedure, we performed Western blotting analysis and q-PCR experiments to examine the levels of EGFP-UL76 protein and mRNA transcript under the same experimental conditions (Figure 2B,C and Figure 3B,C). In these two experiments, cells treated with bortezomib 25 nM and MG132 1 μM were used in parallel as proteasome inhibitory controls. We obtained similar results that the ratios of EGFP-UL76/tubulin under treatment with bortezomib and MG132 showed no difference from the control level, which was consistent with a previous report [4].

Compound **1** did not affect protein ratios of EGFP-UL76/tubulin proteins at 1, 5, and 25 μg/mL of any kind. However, for cells treated with compound **2** at 1, 5, and 25 μg/mL, the ratios were 1.00 (*p* = 0.9466), 0.91 (*p* = 0.0359), and 0.88 (*p* < 0.0010), respectively (Figure 2B and Figure 3B). Nevertheless, the cytotoxic ED_50_ for compound **2** was greater than 25 μg/mL for HEK293T cell in both assays (Appendix A). The protein ratios for compound **3** were 0.91 (*p* < 0.0010), 0.96 (*p* = 0.1484), and 0.86 (*p* < 0.0010), respectively. The protein ratios for compound **4** were 0.92 (*p* = 0.0045), 0.95 (*p* = 0.0420), and 0.86 (*p* = 0.0014), respectively. However, HEK293T cells treated with compound **3** and **4** at 25 μg/mL exhibited a lighter toxicity with ED_50_ values of 19.56 and 21.33 μg/mL, respectively. Compound **5** exhibited significant reduction at 5 and 25 μg/mL; the ratios were 0.78 (*p* < 0.0010) and 0.15 (*p* < 0.0010), respectively, which is consistent with cytotoxicity ED_50_ values (Appendix A). Compound **6** did not affect protein ratios at any tested concentrations. The results from both the quantitative PCR and Western blotting analyses revealed that in the tested compound treatment neither the protein nor the mRNA ratios for EGFP-UL76/GADPH were elevated (Figure 2C and Figure 3C). Taking all these results in account, we suggested that the increase in EGFP-UL76 high-content measurement was likely due to the modulation of protein conformation.

After the results, we investigated the phenotypic size of aggresomes and analyzed the high-content data from Figure 2 and Figure 3 by diameter with methods described previously [4]. As shown in the top panels of Figure 4 and Figure 5, compounds **1**, **2**, **3**, **4**, **5**, and **6** exhibited the highest ratio increases for count, which were as follows: for pit aggresomes: 1.18 (*p* = 0.0331), 1.73 (*p* < 0.0010), 1.76 (*p* < 0.0010), 1.48 (*p* < 0.0010), 1.29 (*p* = 0.0236), and 1.32 (*p* = 0.0062), respectively; for vesicle aggresomes, 1.43 (*p* = 0.0028), 1.88 (*p* < 0.0010), 1.84 (*p* < 0.0010), 1.54 (*p* < 0.0010), 1.53 (*p* < 0.0010), and 1.37 (*p* = 0.0071), respectively. Similar profiles were observed for the ratios of integrated intensity per cell (Figure 4 and Figure 5, middle panels). Compounds **1**, **2**, **3**, **4**, **5**, and **6** showed the highest ratio increases which were as follows: for pit aggresomes, 1.18 (*p* = 0.0130), 1.48 (*p* < 0.0010), 1.31 (*p* = 0.0055), 1.33 (*p* < 0.0010), 1.41 (*p* < 0.0010), and 1.21 (*p* = 0.0097), respectively; for vesicle aggresomes, 1.16 (*p* = 0.0220), 1.73 (*p* < 0.0010), 1.76 (*p* < 0.0010), 1.05 (*p* = 0.5352), 1.29 (*p* = 0.0236), and 1.32 (*p* =0.0063), respectively. The ratios of average intensity per cell were the same for pit and vesicle aggresomes (Figure 4 and Figure 5, bottom panels). For compounds **1**, **2**, **3**, **4**, **5**, and **6** showed the highest increases in ratios observed which were as follows: 1.27 (*p* = 0.0099), 1.42 (*p* < 0.0010), 1.25 (*p* = 0.0034), 1.22 (*p* = 0.0628), 1.42 (*p* < 0.0010), and 1.19 (*p* = 0.0135), respectively.

## 3. Discussion

During the HCS we assessed dolabellanes-based clavinflol C (**1**), stolonidiol (**2**), and stolonidiol-17-acetate (**3**) and clavinflol B (**4**) with proteasome inhibition activities. Several groups demonstrated that stolonidiol (**2**) and stolonidiol-17-acetate (**3**) enhance the activities of choline acetyl transferase (ChAT) [8], which likely is mediated by protein kinase C [9]. ChAT catalyzes the production of acetylcholine which is an essential neurotransmitter. Moreover, proteasome inhibitor MG132 stabilizes ChAT steady-state protein levels and increases the enzyme activity [10]. Therefore, we reason that stolonidiol may mediate the elevation of ChAT activity via both pathways in the inhibition of proteasome and the activation of protein kinase C [9,10,11]. Among the four dolabellanes-based compounds with stolonidiol (**2**) 5 μg/mL treatment, the EGFP-UL76 displayed the highest increase ratios of 2.12 and 1.75 of integrated and average intensity, respectively (Figure 2). Consistent to this result, the tolonidiol (**2**) potentiated the highest ratios of aggresome pit and vesicle integrated and average intensities (Figure 4). For the remaining three, increased ratios were moderate, greater than 10% increase, for all the high-content measurements. Compared to compounds **1** and **4**, compound **2** and **3** show a lack of chloride at C-6, which may contribute to a less cytotoxic effect for HEK293T cell and increased efficacy for proteasome inhibition. Acetylation at C-17 of compound **3** may reduce activity in proteasome inhibition.

Overviewing the known proteasome inhibitors with steroid structure were polyhydroxylated sterol-based agosterols [12] and secosteroid-based physalin B and C [13,14]. Endogenous 25-hydroxyvitamin D (25OHDO) was identified to impair sterol regulatory element-binding proteins (SREBPs) activation by inducing proteolytic processing and ubiquitin-mediated degradation of SREBP cleavage-activating protein (SCAP) [15]. In this study, secosteroid-based compound **6** showed the highest inhibitory measurements of EGFP-UL76 integrated and average intensities and at 1 μg/mL (Figure 3), whereas with 5 μg/mL treatment the EGFP-UL76 exhibited the highest increase ratios of pit and vesicle measurements (Figure 5). As for the secosteroid related compound **5**, it showed less potent of proteasome inhibition with 25 μg/mL treatment which exhibited comparable activities to those of compound **6**. Compound **6** with a conjugated double bond may contribute cytotoxicity to HEK293T cells and display a significant reduction of all high-content measurements at 25 μg/mL.

## 4. Materials and Methods

### 4.1. Compounds ***1***–***6***

In the present study, compounds **1**–**6** (Figure 1) were isolated from Formosan soft coral *Clavularia flava* (wet weight of 1.1 kg), which was collected at Green Island, Taiwan. Briefly, corals were minced and exhaustively extracted with acetone. The organic extract was partitioned between ethyl acetate and water, and the ethyl acetate layer was separated further over a normal phase silica gel by column chromatography eluted with *n*-hexane, ethyl acetate, and methanol to yield 15 fractions. Fraction 14 (116.5 mg) eluted with *n*-hexane–EtOAc (1:5) was subjected to a RP-18 gravity column (MeOH/H_2_O, 70:30 to 100% MeOH) to separate 6 subfractions. Subsequently, a subfraction 14-3 (23 mg) was purified by RP-18 HPLC (50% acetonitrile in H_2_O) to obtain 1 (10.2 mg) and 2 (3.2 mg). Fraction 13 (96.7 mg) eluted with *n*-hexane–EtOAc (1:3) was subjected to column chromatography on silica gel using *n*-hexane–EtOAc gradient (10:1 to 1:10) for elution to give 11 subfractions. Subsequently, a subfraction 13-6 (22 mg) was purified by RP-18 HPLC (60% MeOH in H_2_O) to obtain 3 (5.2 mg). Fraction 15 (86.5 mg) eluted with *n*-hexane–EtOAc (1:10) was subjected to a RP-18 gravity column (MeOH/H_2_O, 70:30 to 100% MeOH) to separate 7 subfractions. Subsequently, a subfraction 15-4 (20 mg) was purified by RP-18 HPLC (50% ACN in H_2_O) to obtain 4 (2.4 mg), **5** (3.2 mg), and **6** (3.0 mg).

### 4.2. High-Content Screening (HCS) Assay for Proteasome Inhibition

We followed the HCS assay routinely conducted in the lab. In brief, the complete assay includes a series of experiments [4]. Experiments were sequentially performed: (1) the DNA transfection into cell culture with compound treatment, (2) cells fixation, (3) image acquisition using an ImageXpress Micro Widefield HCS system (Molecular Device, San Jose, CA, USA), (4) high-content measurements analyzed by modules of MetaExpress, Cell Scoring and Multi-Wavelength Cell Scoring, (5) Western blotting imaging and densitometric analysis, and (6) RNA purification and quantitative PCR.

## Figures and Tables

**Figure 1 marinedrugs-18-00039-f001:**
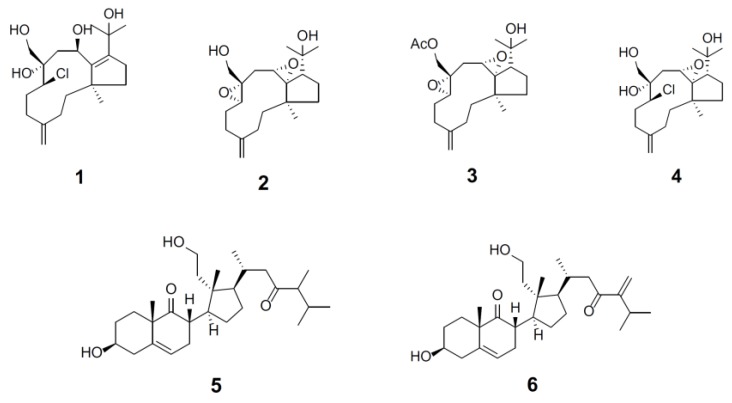
Marine natural products exhibit proteasome inhibition by high-content assays. Dolabellanes are clavinflol C (**1**), stolonidiol (**2**), stolonidiol-17-acetate (**3**), and clavinflol B (**4**). Secosteroids are 3*β*,11-dihydroxy-24-methyl-9,11-secocholest-5-en-9,23-dione (**5**) and 3*β*,11-dihydroxy-24-methylene-9,11-secocholest-5-en-9,23-dione (**6**).

**Figure 2 marinedrugs-18-00039-f002:**
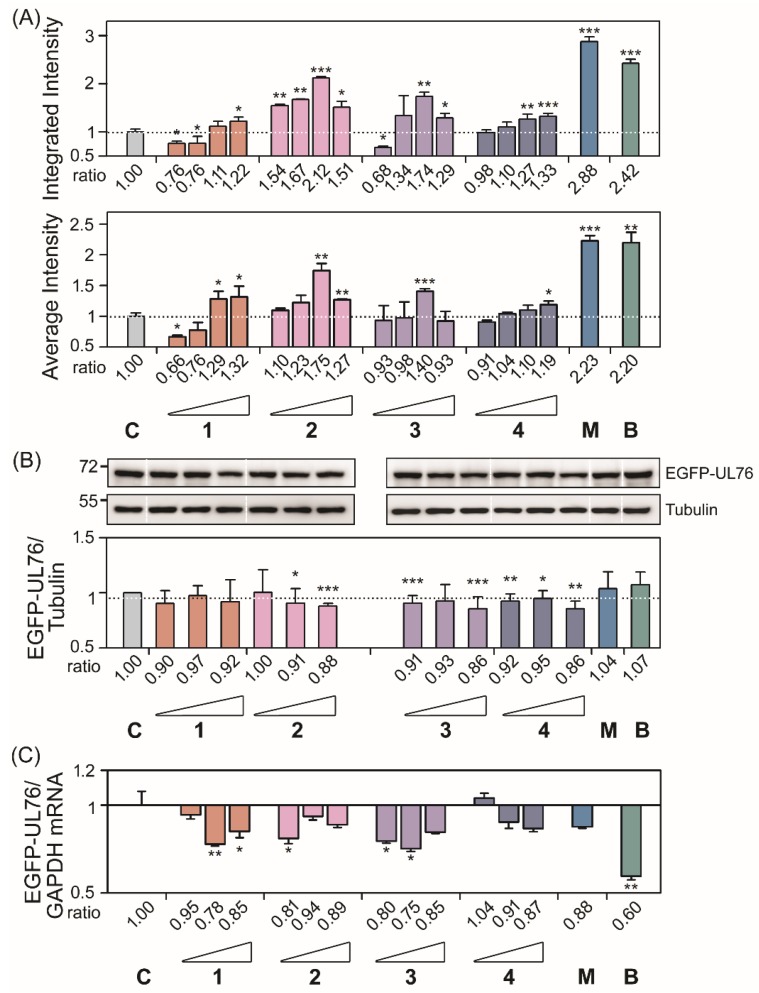
The assessment of proteasome inhibitory activity of marine dolabellanes-based compounds (**1**, **2**, **3**, and **4**) using a standard operation protocol of high-content EGFP-UL76 aggresomes screening assay. Pure compounds modulated high-content measurements of EGFP-UL76 aggresomes. (**A**) Assessment of the integrated and average intensities of EGFP-UL76 aggresomes (1 to 50 μm) per cell. The tested concentrations were 0.2, 1, 5, and 25 μg/mL for pure compounds **1**, **2**, **3**, and **4**, respectively. The integrated (top panel) and average (bottom panel) intensities per cell were measured, and the ratios were obtained by normalization to the control without proteasome inhibitor treatment, which is denoted by **C** throughout the text. (**B**) Validation of the EGFP-UL76 protein levels upon the addition of the tested marine compounds. Western blot imaging and densitometric analyses were performed to quantitate the EGFPUL76/tubulin protein ratio with the addition of tested compound treatment at 1, 5, and 25 μg/mL. The molecular mass markers are shown on the left in kDa. (**C**) Quantitative PCR was conducted to assess the transcript ratio of EGFP-UL76/GAPDH in HEK293T cells treated with pure compounds at 1, 5, and 25 μg/mL. The high-content measurements of EGFP-UL76 with the addition of the proteasome inhibitors, bortezomib (25 nM, denoted **B**) and MG132 (1 μM, denoted **M**), were used as positive controls. All data points are the averages of at least three repetitive experiments. The error bars indicate standard deviations. The following symbols are used to indicate statistical significance throughout the text: * 0.01 < *p* < 0.05; ** 0.001 < *p* < 0.01; *** *p* < 0.001.

**Figure 3 marinedrugs-18-00039-f003:**
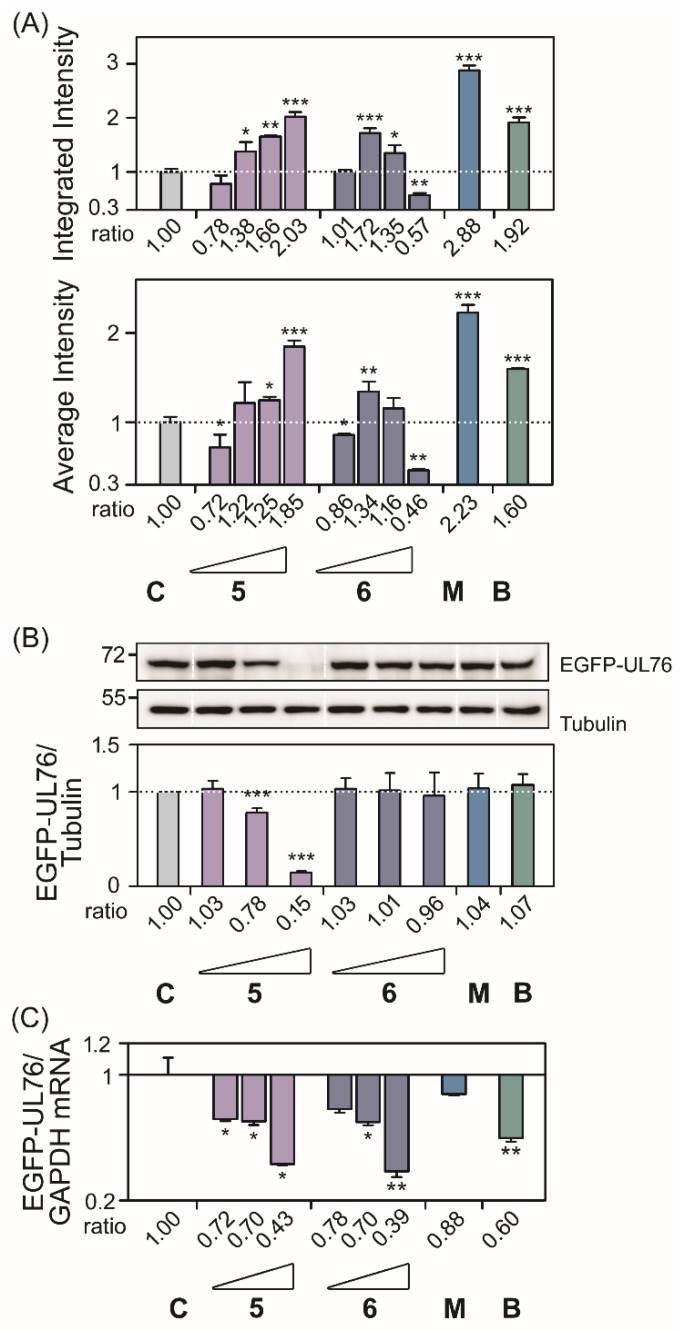
The assessment of proteasome inhibitory activity of marine secosteroid-based compounds (**5** and **6**) using a standard operation protocol of high-content EGFP-UL76 aggresomes screening assay as described in Figure 2 legend.

**Figure 4 marinedrugs-18-00039-f004:**
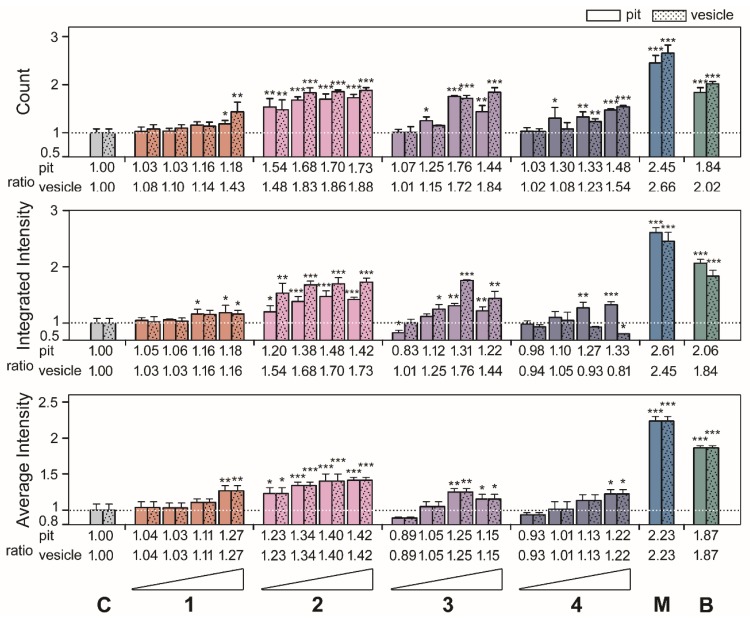
Classification of the high-content measurements of EGFP-UL76 aggresomes by size with marine dolabellanes (**1**, **2**, **3**, and **4**) at 0.2, 1, 5, and 25 μg/mL treatment. Pit and vesicle denote aggresomes 1 to 5 μm and 5 to 20 μm in diameter, respectively. Measurements of pit and vesicle aggresomes per cell were as follows: count number, integrated intensity, and average intensity. The relative ratio was normalized to control values without pure compound treatment. All data points are the averages of at least three repetitive experiments. The error bars indicate standard deviations. The following symbols are used to indicate statistical significance throughout the text: * 0.01 < *p* < 0.05; ** 0.001 < *p* < 0.01; *** *p* < 0.001.

**Figure 5 marinedrugs-18-00039-f005:**
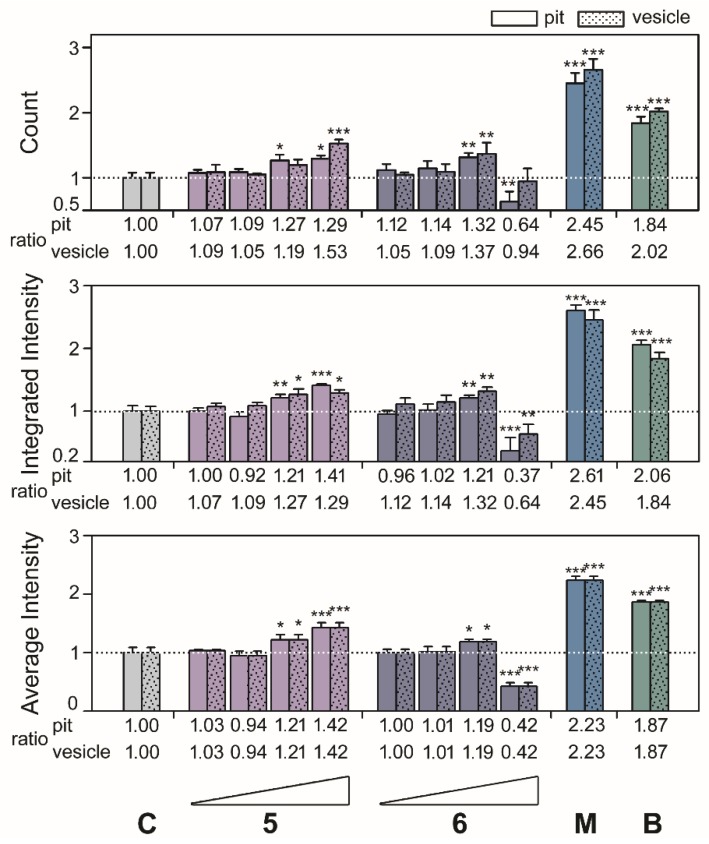
Classification of the high-content measurements of EGFP-UL76 aggresomes by size with marine secosteroid-based compounds (**5** and **6**). Treatment as described in Figure 4 legend.

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
