# Peer review of "Ubiquitin-Proteasome Modulating Dolabellanes and Secosteroids from Soft Coral Clavularia flava"

_marinedrugs, 2020, doi:10.3390/md18010039_

Round 1

Reviewer 1 Report

The paper with the title

Ubiquitin-Proteosome Modulating Dolabellanes and 2 Secosteroids from Soft Coral Clavularia Flava submitted by:

Che-Yen Chiu1†, Xue-Hua Ling 1†, Shang-Kwei Wang 2,*, and Chang-Yih Duh 1,*

describes the purification and  identification of a novel substances from Soft Coral, and effect of these substances on proteasome activity and cell viability.

The substances were obtained by chromatographic fractionation of ethyl acetate solubles from C. flava and identified by comparison of their spectral data with those of reported literatures or by analysis of the spectral data.

The substance were tested for their ability to inhibit the proteasome by a published assay and effects on cell viability were analysed by a commercial MTT assay.

The methods used and described by the authors are well suited and state of the art for the aim of the experiments. The description of the Materials and methods is sufficient for reproduction. The introduction as well as the cited literature are appropriate.

The inhibitory effects of the reported substances are not very impressive, but clearly detectable.

The discussion is extremely short and rather a summary of the results.

The authors should compare the newly described substances with known inhibitors with respect to proteasome inhibition/ cyto toxicity ratio in the discussion and not with a table at the end of the supplementary information. Are their strategies possible to use such substances as lead compounds for optimization by substitution of side chains ?

Lane 217 starting 6. apparently contains an comment from a referee which was probably  accidently copied to the text.

Author Response

Reviewer #1:

Comparison of the activity was added at the end of each paragraphs of the discussion section. A description of the activity was added to the abstract. Lane 217 (starting 6) to lane 220 were deleted.

Reviewer 2 Report

The purpose of this paper is to discover bioactive compound with proteasome inhibitory activity. Indeed, authors were succeeded to isolate 6 molecules including 3 new compounds.

If this report is to be published in Marine Drugs, it would be better to modify about followings.

The abstract contains only information on the isolated compound, and it is necessary to add a description of the latter half about the activity such as the level of it. Determining the structure of compound 1 can be expected to be difficult. The authors presume the structure of 1 because it is similar to known compound 2~4. Although the proposed structure seems reasonable, the stereochemistry of C10-OH remains questionable. C10-OH exists in an 11-membered ring, and its conformation is not fixed. As a result, even if C10-OH is in the α configuration, it seems possible to explain the underlying NOESY correlations. In addition, it is considered that 12-H of compound 4 was abstracted, and the epoxide was cleaved to generate 1, and even in this case, the configuration would be α configuration.

The structure of Compound 5 is determined by comparison with Reference 7. The sentence says, "The relative stereochemistry at ~ were found to be the same as those of ~." I felt that the explanation should be added due to the lack of grounds.

In the discussion on pages 7 to 8 merely describes the results of the activity test. I think additional conclusions or further considerations will be more informative to the reader.

Minor points

2 L70 C28H46O4 => C28H46NaO4 2 L70 1H1-H COSY =>  1H-1H COSY 6 footnote @ => µ 8 L217~220 Authors should~. => delete 8 L228, L234 to to => to

Reference 1, page number 1370 was lacked

Reference 4, page number 395 was lacked

Author Response

Reviewer #2:

A description of the activity was added to the abstract. If C10-OH is in the α configuration, it is not possible to explain the NOESY correlations from H-10 to H3-15 and from H-10 to H-7. The explanation of determination of relative stereochemistry by NOESY experiment was added. Conclusions were added at the end of each paragraphs. Page 2, L70, C28H46O4 was changed to C28H46NaO4. Page 2, L70, 1H1-H COSY was changed to 1H-1H COSY. Page 6, footnote @ was changed to µ. Page 8, L228, L234 ‘to to’ was changed to ‘to’. Reference 1, paper number 1370 was added Reference 4, paper number 395 was added
